# Facile Controlled Synthesis of Pd-ZnO Nanostructures for Nitrite Detection

**DOI:** 10.3390/molecules28010099

**Published:** 2022-12-23

**Authors:** Yaojuan Hu, Fengyun He, Changyun Chen, Changli Zhang, Jingliang Liu

**Affiliations:** School of Environmental Science, Nanjing Xiaozhuang University, Nanjing 211171, China

**Keywords:** Pd-ZnO nanostructures, nitrite oxidation reaction, effect of surface structure, nitrite sensor

## Abstract

The electrocatalytic characteristics of nanostructures are significantly affected by surface structure. The strict regulation of structural characteristics is highly beneficial for the creation of novel nanocatalysts with enhanced electrocatalytic performance. This work reports a nitrite electrochemical sensor based on novel flower-like Pd-ZnO nanostructures. The Pd-ZnO nanocatalysts were synthesized through a simple hydrothermal method, and their morphology and structure were characterized via field-emission scanning electron microscopy (FE-SEM), X-ray photoelectron spectroscopy (XPS), and X-ray diffraction (XRD). Their electrocatalytical performance in the nitrite oxidation reaction was studied via cyclic voltammetry (CV) and the amperometric technique. Compared to pure ZnO and Pd nanoparticles, the Pd-ZnO nanostructures exhibited enhanced electrochemical performance in the nitrite oxidation reaction. In order to investigate the relationships between the structures of Pd-ZnO nanocatalysts and the corresponding electrocatalytic performances, different surface morphologies of Pd-ZnO nanocatalysts were fabricated by altering the solution pH. It was found that the flower-like Pd-ZnO nanostructures possessed larger effective surface areas and faster electron transfer rates, resulting in the highest electrocatalytic performance in the nitrite oxidation reaction. The designed nitrite sensor based on flower-like Pd-ZnO displayed a wide concentration linear range of 1 µM–2350 µM, a low detection limit of 0.2 µM (S/N of 3), and high sensitivity of 151.9 µA mM^−1^ cm^−2^. Furthermore, the proposed sensor exhibited perfect selectivity, excellent reproducibility, and long-time stability, as well as good performance in real sample detection.

## 1. Introduction

As food additives, nitrite ions (NO_2_^−^), which act as meat preservatives, coloring agents, artificial flavor enhancers, and so on, usually exist in processed foods [1]. However, the excessive adsorption of nitrite can pose a great threat to human health, because it can cause the irreversible transformation of hemoglobin to methemoglobin and reduce the ability of blood to transport oxygen [2,3]. The World Health Organization (WHO) has reported that nitrite can damage the spleen, kidneys, and nervous system. Thus, to guarantee public health, the nitrite levels in a wide variety of foods are strictly regulated in many countries. For example, the maximum allowance of nitrite in dry-cured meat products is 30 mg/kg in China and 50–250 mg/kg in the European Union. Therefore, the development of an accurate, rapid, and economical method for the detection of the concentration of nitrite is of great significance in food safety.

To date, diverse analytical methods have been applied for nitrite detection, such as spectrophotometry [4], chromatography [5], chemiluminescence [6], and surface-enhanced Raman scattering spectroscopy (SERS) [7]. However, most of these techniques involve complicated processes and are time-consuming. Owing to their high accuracy, rapid response, and simple operation, electrochemical methods are currently being developed for the detection of nitrite [8,9,10,11,12]. Generally, electrochemical techniques result from either the anodic oxidation or cathodic reduction of nitrite on the surfaces of different electrodes [13,14,15,16,17,18]. A limitation of the cathodic reduction of nitrite is interference from a few products, which depends on the properties of the electrodes and electrocatalysts [19,20]. Compared with the nitrite reduction reaction, the oxidation of nitrite is often favored since nitrate (NO_3_^−^) is the ultimate product, which is obtained without interference [13,15,21,22,23]. Bare glassy carbon is also highly effective in the direct oxidation of nitrite. However, the electro-oxidation of nitrite on unmodified electrodes always occurs at quite high potential, and the selectivity is poor in the presence of other oxidizing products, thus decreasing the electrode’s sensitivity and accuracy [14,21,24,25,26]. Therefore, it is essential to develop new electrode-modified materials which have enriched active surface areas and can decrease the overpotential to obtain excellent electrochemical performance for the oxidation of nitrite.

Nanoscale noble metals, due to their large specific surface areas and distinctive physicochemical properties, usually act as electrocatalysts for the construction of electrochemical sensors [27,28]. However, the morphologies of noble metal nanocatalysts change during catalytic reactions, which can lead to the agglomeration of nanocatalysts and dramatic decreases in their activity and selectivity. The use of supported nanoscale noble metals has been demonstrated to be the most efficient method to reduce the agglomeration of noble metal nanoparticles and maintain or enhance their activity and stability [23,24,25,26,29,30]. In recent years, nanostructured zinc oxide (ZnO) has been used as a catalyst support due to its lower cost, nontoxicity, and easy-to-prepare morphology. Moreover, ZnO nanostructure supports can induce strong metal–support interaction, which provides the opportunities for adjusting materials’ properties to produce specific sites which affect their catalytic performance [22,31,32,33,34].

In this work, ZnO-supported Pd nanocomposites (Pd-ZnO) were fabricated and used for the construction of a nitrite electrochemical sensor. Because of the combination of the excellent catalytic activity of Pd nanoparticles (Pd NPs) and the large active surface area of ZnO supports, the Pd-ZnO nanocatalysts exhibited higher electrocatalytic activity toward the nitrite. Moreover, it has been reported that the surface structure of nanocatalysts has great influence on their electrocatalytic activities [32,35]. Therefore, different surface structures of Pd-ZnO nanocatalysts were synthesized by altering the solution pH, and their electrocatalytic performance was investigated. Furthermore, the designed sensor exhibited excellent performance for the detection of nitrite and was used for detection in real samples (sausage and pickles). These results indicated that the constructed nitrite sensor has the potential to be used for the detection of nitrite in food products and other real samples.

## 2. Results and Discussion

### 2.1. Synthesis and Characterization of Pd-ZnO Nanocatalysts

The ZnO, Pd, and Pd-ZnO nanocatalysts were fabricated through a simple hydrothermal method with Pd(acac)_2_ and Zn(acac)_2_ as the precursors and PVP and DMF as the stabilizing agent and the reduction agent, respectively. To explore the role of DMF in the formation of composites, we carried out control experiments without DMF. With sufficient DMF, as the reaction proceeded, the color of the mixture changed from ivory to gray-black, and after washing and drying, a black solid product was obtained. On the contrary, without DMF, we discovered that the mixture remained ivory, and there was no black solid product obtained; we therefore concluded that DMF was the reduction agent.

Figure 1 shows typical SEM images of ZnO, Pd, and Pd-ZnO nanostructures. It can be seen from Figure 1 that the ZnO (Figure 1a), Pd (Figure 1c), and Pd-ZnO (Figure 1e) nanostructures were fabricated at a large scale and are well dispersed. Moreover, the fabricated ZnO nanocatalysts had flower-like structures with sizes of 700-800 nm (Figure 1b). However, when the precursor was Pd(acac)_2_ alone, the prepared Pd nanostructures were small spherical nanoparticles (Figure 1d). When the reaction was performed in the presence of Pd(acac)_2_ and Zn(acac)_2_, the resulting Pd-ZnO nanocomposites also exhibited flower-like structures like the ZnO nanostructures (Figure 1f). Figure 1g shows an HAADF-STEM image of flowerlike Pd-ZnO nanocomposites, on which elemental mapping analysis was carried out. Figure 1h–j show the energy-dispersive X-ray spectroscopy (EDS) elemental mapping images of Zn, O, and Pd, respectively. The elemental mapping analysis clearly revealed that the support was mainly composed of Zn and O, and Pd NPs on the surface of ZnO were uniformly dispersed. 

In order to investigate the properties of the catalyst surfaces, XPS analysis was carried out to explore the distribution and chemical states in the Pd-ZnO nanostructures. The surveyed XPS spectrum of the Pd-ZnO nanostructures is displayed in Figure 2A. The peaks of Zn, O, and Pd confirm that the Pd NPs were successfully anchored on the surfaces of the ZnO nanostructures. The core level Zn 2p XPS spectrum of the Pd-ZnO nanostructures shows two binding energy peaks at 1021.8 eV and 1044.9 eV (Figure 2B), which generally corresponded to Zn 2p_3/2_ and Zn 2p_1/2_, respectively [33]. The binding energy difference between the Zn 2p_3/2_ and Zn 2p_1/2_ peaks was about 23 eV, demonstrating the presence of Zn^2+^ in the Pd-ZnO nanostructures [36,37]. The 3d core level XPS spectrum consisted of two peaks for metallic palladium at 335.0 eV (Pd 3d_5/2_) and 340.2 eV (Pd3d_3/2_), with two more small peaks at 336.4 eV and 341.5 eV, which were attributed to the existence of Pd^2+^ [14,33]. The relative intensities of those components (Pd^0^ and Pd^2+^) indicated that Pd in the Pd-ZnO nanocatalysts was predominately Pd^0^, which could provide more suitable sites for nitrite oxidation than Pd^2+^ [38].

The crystal structures of the ZnO, Pd, and Pd-ZnO nanostructures were confirmed via XRD measurements (Figure 2D). The patterns of the Pd-ZnO nanostructures exhibited all of the major peaks of ZnO at 31.7° (100), 34.4° (002), 36.2° (101), 47.5° (102), 56.6° (110), 62.8° (103), 68.0° (112), and 69.1° (201) [32]. The peak at ca. 40.3° came from the pattern of Pd (111), and the little shift in the peak compared with pure metallic Pd was probably caused by the interaction between metallic Pd and ZnO [37]. The XRD patterns of ZnO nanostructures could be indexed to typical wurtzite structures (JCPDS 36-1451), the XRD patterns of Pd could be indexed to typical face-centered-cubic (fcc) structures of Pd (JCPDS 46-1043), and no other crystalline structures were detected.

To examine the electrochemical properties of the ZnO, Pd, and Pd-ZnO nanocatalysts, electrochemical characterizations of the modified electrodes were performed via cyclic voltammetry (CV) and electrochemical impedance spectroscopy (EIS) in Fe(CN)_6_^3−^/^4−^ solution (Appendix A). Appendix A depicts the typical CVs of ZnO/GCE (curve a), Pd/GCE (curve b), and Pd-ZnO/GCE (curve c). A pair of well-defined redox peaks could be observed at each modified electrode. However, compared with ZnO/GCE and Pd/GCE, Pd-ZnO/GCE showed a higher peak current and a lower peak separation (Δ*Ep*) (curve c), which demonstrated the effective increase in the electron transfer rate of the Fe(CN)_6_^3−^/^4−^ redox reaction due to the presence of Pd NPs on the Pd-ZnO/GCE [2,39]. On the other hand, the increased current in the Pd-ZnO/GCE was due to the presence of ZnO, the development of an effective surface area, and a higher level of attraction of Fe(CN)_6_^3−^/^4−^ on the electrode surface [31]. The EIS technique is a helpful tool to probe electron transfer kinetics. Appendix A exhibits the Nyquist plots of different modified electrodes using the Fe(CN)_6_^3−^/^4−^ redox probe. As can be seen, the semicircle diameter of Pd-ZnO/GCE (curve c) was smaller than that of ZnO/GCE (curve a) and Pd/GCE (curve b), which implies that the Pd-ZnO nanocomposites exhibited the fastest electrochemical reaction kinetics.

### 2.2. Electrocatalytic Properties of Pd-ZnO /GCE

The electrochemical responses of NO_2_^-^ at ZnO/GCE, Pd/GCE, and Pd-ZnO/GCE were investigated in PBS solution (pH 7.0), and their electrocatalytic performances were compared. As shown in Figure 3, no distinct peaks appeared in blank PBS (curve a in panel A, B, and C). Upon the addition of nitrite (0.5 mM), obvious anodic peaks attributed to the oxidation of nitrite appeared at ca. 0.88 V for ZnO/GCE (panel A), ca. 0.86 V for Pd/GCE (panel B), and ca. 0.78 V for Pd-ZnO/GCE (panel C). Moreover, the nitrite oxidation peak potential at Pd-ZnO/GCE (0.78 V) was more negative than that of other electrode materials, such as RGO/MnFe_2_O_4_/polyaniline at 0.86 V [39], AuNPs/CP at 0.80 V [3], and CDs-Au-N at 0.83 V [25], suggesting the significantly high catalytic activity and selectively of the Pd-ZnO nanostructures toward nitrite oxidation. Furthermore, the peak current of the nitrite oxidation was in the order of Pd-ZnO/GCE > Pd/GCE > ZnO/GCE. The highest electrocatalytic performance of the Pd-ZnO nanocatalyst could be ascribed to the combination of two factors: one is the excellent electrocatalytic activity of Pd NPs, and the other is the large electrochemical active surface area of the ZnO support, which could promote the absorption of more nitrite ions and thus enhance the current response [31].

### 2.3. Effects of Surface Structure

To investigate the effect of nanocatalysts’ structures on their electrocatalytic properties, different surface morphologies of Pd-ZnO nanocatalyst were fabricated by altering the solution pH, which could be adjusted by adding different volume of NaOH, and the electrocatalytic activities of Pd-ZnO nanocatalysts with different structures were studied. Typical SEM images are shown in Figure 4. Panel a and b show the SEM images of the synthesized piece-like Pd-ZnO nanocomposites, which were created without NaOH, and the pH of the reaction solution was ca. 6. The diameter and thickness of the piece-like Pd-ZnO was ca. 15 nm and ca. 8 nm, respectively, and the surface of the Pd-ZnO was smooth, which implies that the surface area of this type of nanocatalyst was small. When 500 µL of 0.25 M NaOH solution was added, the pH of the reaction solution was increased to 7, and tube-like Pd-ZnO nanocomposites were synthesized (Figure 4c). The length of these tubes varied from 400 to 500 nm, with diameters from 50 to 100 nm (Figure 4d). When the volume of NaOH solution increased to 1000 µL, the pH of the reaction solution was increased to 8; the tubes tended to become slim rods, and many long and thin branches overlapped together. Finally, flower-like Pd-ZnO were obtained (Figure 4e,f). At the NaOH volume of 2000 µL and pH ≈ 9, the branches of the flower-like Pd-ZnO became slighter and more branchy, which meant they could exhibit larger active surface areas (Figure 4g,h). If the NaOH volume further increased to 2500 µL and the reaction pH value exceeded 10, the flower-like Pd-ZnO nanostructures were destroyed, and the nanocatalysts appeared as disorder structures, which were composed of small nanosheets (Figure 4i and j) and implied a decrease in the active surface area. 

In order to understand the effect of pH on the morphologies of the nanocomposites, we explain the synthetic mechanism below. First, without using NaOH, the precursor of the zinc source was first decomposed to Zn^2+^, and a large amount of Zn^2+^ was generated into an oversaturated state, followed by the nucleation and growth of ZnO. Therefore, Zn^2+^ was the “growing unit” for ZnO nanocrystals under the low-alkalinity condition (pH ≈ 6), which finally led to the formation of piece-like shapes. When the NaOH solution (500 µL, 0.25 M) was added to the reaction solution (pH ≈ 7), OH^−^ initially reacted with Zn^2+^ and formed Zn(OH)_2_, and during hydrolysis at 140 °C, the Zn(OH)_2_ colloid partially dissolved into Zn^2+^ and OH^-^, cresting the ZnO nuclei. The other Zn(OH)_2_ further reacted with OH^−^ and formed [Zn(OH)_4_]^2−^. Thus, [Zn(OH)_4_]^2−^ grew along the (002) direction of the ZnO crystal nuclei and formed tube-like Pd-ZnO nanocomposites. As the concentration of OH^−^ (pH 8–9) increased, more [Zn(OH)_4_]^2−^ were formed and grew along the (002) direction of the ZnO crystals, many slim and long ZnO nanorods formed and overlapped together, and the flower-like composites were generated. At pH ≈ 10, owing to the high concentration of OH^−^, excessive negatively charged [Zn(OH)_4_]^2−^ were formed, and the growth in other directions became comparatively faster than that in the (002) direction; thus, the nanocatalysts appeared to have disorder structures. 

Furthermore, the specific surface areas of these Pd-ZnO nanocatalysts were further characterized via nitrogen adsorption analyses. As can be seen from the nitrogen adsorption–desorption isotherm plots (Appendix A), these Pd-ZnO composites exhibited different specific surface areas (16–45 m^2^ g^−1^), and the flower-like Pd-ZnO exhibited the highest specific surface area and thus enhanced the electrocatalytic activity.

Then, the effects of nanocatalysts’ morphologies on the electrocatalytic activities were studied via CVs (Appendix A). As exhibited in Appendix A, Pd-ZnO nanocatalysts with different morphologies exhibited different electrocatalytic characteristics for nitrite oxidation. The catalytic current increased with increase in the NaOH volume added, and the maximum value was attained at the NaOH volume of 2000 µL (curve a–d). Moreover, the anodic peak potentials of different Pd-ZnO nanocatalysts were different, and the flower-like Pd-ZnO nanocatalysts exhibited the most negative peak potential. The results indicated that the morphologies of Pd-ZnO play a crucial role in their electrocatalytic activities. This phenomenon could be ascribed to the significant differences between the electrochemically active areas in Pd-ZnO fabricated with different NaOH volumes. The flower-like Pd-ZnO nanostructures with more slender branches, which were synthesized in 2000 µL of NaOH, exhibited the largest active surface areas and were most favorable for mass transfer. Thus, the flower-like Pd-ZnO nanostructures with more slender branches were chosen for further investigation. 

### 2.4. Effect of Scan Rate

To understand the kinetics mechanism of the modified electrode reaction, we studied the relation between the peak potential, the peak current of nitrite oxidation, and different scan rates. Appendix A exhibits the typical CVs of the Pd-ZnO/GCE in PBS (pH 7.0) containing 0.5 mM nitrite at different scan rates (10–500 mV s^−1^). As shown in Appendix A, the peak current (*i*_p_) increased proportionally with the square root of scan rate (*v*^1/2^). Thus, the oxidation of nitrite on the surface of the modified electrode was predominantly a diffusion-controlled process [17]. Appendix A shows that the peak potential (*Ep*) had a proportional relationship with the natural logarithm of scan rate (ln *v*), which can be expressed as *Ep* = 0.7061 + 0.034 ln*v* (R^2^ = 0.9982). For an irreversible electrode process, the *Ep* could be determined by the Laviron equation:(1)$$E_{p}=E^{0\prime }+\left(\frac{RT}{\alpha nF}\right)\ln\left(\frac{RT\kappa^{0}}{\alpha nF}\right)+\left(\frac{RT}{\alpha nF}\right)lnv$$
where $E^{0\prime }$ is the standard electrode potential (V), *α* is the electron transfer coefficient, $\kappa^{0}$ is the heterogeneous electron transfer rate constant (s^−1^), and the other terms have their usual conventional meanings. The number of transfer electrons (*n*) in nitrite electrochemical oxidation was calculated to be about 2. Therefore, the nitrite oxidation at the modified electrode surface was a two-electron transfer process, which was consistent with the reported literature [23,26].

### 2.5. Effect of the Solution pH

The effect of the solution’s pH on the electrochemical response of nitrite in Pd-ZnO/GCE was recorded in the pH range from 4.0 to 9.0. As shown in Appendix A, the peak potential remained nearly unchanged with the variation in the solution’s pH; however, the peak current showed significant variation with the variation in the solution’s pH. The variation in the peak current with the pH is plotted and shown in Appendix A. It can be seen that the anodic peak current was enhanced with the pH value from 4.0 to 7.0 and reached a maximum at pH 7.0, and a further increase led to a decrease in the oxidation peak current. A similar phenomenon was also obtained in other studies in the literature [13,21,39], which may be ascribed to the following reasons: (1) NO_2_^−^ are instable when the pH is small, which could be shown as the following equation [10,11,12]:(2)$$2H^{+}+3NO_{2}^{-}⇄2NO+NO_{3}^{-}+H_{2}O$$

(2) The electrooxidation of NO_2_^−^ is a proton-dependent process, and due to the lack of protons, the nitrite oxidation reaction become more difficult at a higher pH (>7.0) [14,15]. Therefore, pH 7.0 was chosen as a supporting electrolyte for further electrochemical experiments.

### 2.6. Electrochemical Detection of Nitrite 

To assess the electrochemical determination of nitrite in Pd-ZnO-modified electrodes, the amperometric technique was applied. Figure 5A presents the typical amperometric response of Pd-ZnO/GCE towards successive injections of nitrite into 0.1 M PBS at 0.78 V. As can be seen, after the addition of nitrite, a rapid increase in the oxidation current was observed and reached the steady-state value within 3 s, suggesting the modified electrode displays a rapid response toward nitrite oxidation, which is desirable for the real-time analysis of real samples. Moreover, Figure 5B shows the corresponding calibration plot generated from Figure 5A. It clearly exhibited a good linear relationship between the oxidation current and the nitrite concentrations from 1 µM to 2350 µM with a correlation coefficient of 0.9994, sensitivity of approximately 151.9 µA mM^−1^ cm^−2^, and the detection limit of ca. 0.2 µM with a signal/noise ratio (S/N) of 3. These analytical performances of the Pd-ZnO-modified electrode are superior or comparable to those previously reported in other studies (Table 1).

The selectivity capability plays a key role in estimating the performance of a developed sensor, because various types of interference may co-exist in practical samples. In order to investigate the selectivity of the developed sensor, the amperometric responses of some possible interferents, including some inorganic ions (e.g., Na^+^, K^+^, Ca^2+^, NO_3_^−^, and CO_3_^2−^) and some electroactive materials (e.g., glucose, uric acid (UA), and ascorbic acid (AA)) along with 0.5 mM nitrite were recorded at different applied potentials and are exhibited in Figure 6A. The concentrations of interferent species were 10-fold that of nitrite. It can be seen that these inorganic ions had no obvious impact on the detection of nitrite. However, for some electroactive materials, especially glucose, there were obvious signals at lower potentials (25% at 0.6 V and 18% at 0.7 V). However, the interference at the applied potential (0.78V) was below 5%, indicating that the fabricated nitrite sensor had the biggest current response at 0.78 V, and the sensor was highly selective toward the detection of nitrite in the presence of these co-existent species.

### 2.7. Reproducibility and Stability

To further evaluate the reproducibility of the Pd-ZnO/GCE-based sensor, five Pd-ZnO-modified electrodes were prepared under identical conditions to analyze the amperometric response of 0.5 mM nitrite. As shown in Figure 6B, the relative standard deviation (RSD) was 3.98%, demonstrating that the Pd-ZnO/GCE sensing platform possesses excellent reproducibility for nitrite detection.

Furthermore, the storage stability of a sensor is one of the critical parameters in practical application. The stability of the sensor was tested by recording the amperometric response toward 0.5 mM nitrite. The results were shown in Figure 6C; after 60 days of storage in ambient conditions, the current response of the sensor remained 92.3% of its initial response, implying that the sensor had excellent long-term stability.

### 2.8. Real Samples Analysis

The practical applications of the fabricated sensor were also evaluated; the modified electrode was used for the detection of the concentration of nitrite in real samples (sausage and pickles). In addition, to confirm the accuracy of the developed sensor, the optical method was used for the measurement of nitrite in the same samples. It is shown that the results detected by the developed sensor were consistent with those obtained using the optical method (Table 2). Recovery testing was performed to verify the validity of the proposed method. The recovery values were between 92.36% and 108.56%. These results suggested that the developed nitrite sensor is suitable for the detection of nitrite in food products and other real samples.

## 3. Experimental Section

### 3.1. Chemical and Materials

Palladium acetylacetonate (Pd(acac)_2_) and zinc acetylacetonate (Zn(acac)_2_) were purchased from Alfa Aesar. Sodium hydroxide (NaOH), polyvinylpyrrolidone (PVP, Mw ≈ 30,000), N,N-dimethylformamide (DMF), and sodium nitrite were all purchased from Sinopharm Chemical Reagent Co. Shanghai, China. All the chemicals were of analytical grade and used as received.

### 3.2. Synthesis of Pd-ZnO Nanocomposites

The Pd-ZnO nanocomposites were synthesized through a simple hydrothermal method [32]. Pd(acac)_2_ and Zn(acac)_2_ were used as precursors, and PVP and DMF were used as the stabilizing agent and reduction agent, respectively. Typically, 1.5 mg of Pd(acac)_2_, 60 mg of Zn(acac)_2_, and 100 mg of PVP were dispersed into a DMF/H_2_O solution (10 mL: 2 mL) under vigorous stirring. In order to adjust the pH of the solution, 2000 µL of NaOH solution (0.25 M) was added into the DMF/H_2_O solution. Then, the resulting solution was placed into a Teflon-lined container and heated at 140 °C for 2 h. After the Teflon-lined container cooled down naturally at room temperature, the obtained powder materials were collected via centrifugation at 10,000 rpm and washed several times with ethanol. Finally, the products were dried at 60 °C in vacuum for further use. In order to study the role of DMF, the same Pd(acac)_2_, Zn(acac)_2_, and PVP were dispersed into H_2_O solution (12 mL) under vigorous stirring, and the other procedures were the same as those above. To investigate the role of ZnO and Pd in the electrocatalytic performance of Pd-ZnO nanocatalysts, the Pd and ZnO nanostructures were also synthesized by using similar procedures as those above. In order to study the effect of the nanocatalysts’ morphologies on the electrocatalytic activities, we also fabricated Pd-ZnO with different surface structures by altering the volume of NaOH, and the volume was 0, 500, 1000, 2000, and 2500 µL.

### 3.3. Apparatus and Procedures

Scanning electron microscopy (SEM) images were recorded with a field-emission scanning electron microscope (FESEM, HITACHI S-4800, Hitachi, Japan ). High-angle annular dark field–scanning transmission electron microscopy (HAADF-STEM) and energy-dispersive X-ray spectroscopy (EDS) maps were created using an FEI Tecanai F20 microscope equipped with an Oxford energy-dispersive X-ray analysis system. The X-ray photoelectron spectroscopy (XPS) was carried out on an ESCALAB 250 XPS spectrometer (VG Scientifics) using the monochromatic Al Kα line at 1486.6 eV. The binding energies were calibrated with respect to the C1s peak at 284.6 eV, and the peak fit analysis was performed using the XPS PEAK program (version 4.0). The X-ray diffraction (XRD) patterns were collected on a Rigaku/Max-3A X-ray diffractometer with Cu Kα radiation (λ = 0.1542 nm).

### 3.4. Electrochemical Measurements

A conventional three-electrode system was used for the electrochemical measurements. A glassy carbon electrode (GCE) modified with different nanostructures was used as a working electrode, a saturated calomel electrode (SCE) as a reference electrode, and a coiled Pt wire as a counter electrode. For the GCE cleaning, the electrode surface was polished with slurries of 0.3 and 0.05 µm alumina sequentially until a mirror surface was achieved. Then, the electrode was rinsed thoroughly with double-distilled water. Under the typical procedure, about 2 mg of the Pd, ZnO, and Pd-ZnO nanostructures was dispersed in 1 mL of deionized water and sonicated. Then, 8 µL of as-prepared suspension was drop casted on the surface of the GCE and dried at ambient temperature.

The electrochemical properties of the modified electrode were studied in 0.1 M KCl solution containing 5 mM K_3_[Fe(CN)_6_]/ K_4_[Fe(CN)_6_]. The cyclic voltammetry (CV) responses were recorded within the potential window −0.1 V to 0.5 V at a scan rate of 50 mV/s. Electrochemical impedance spectroscopy (EIS) was investigated with the frequency range of 10^−2^–10^5^ Hz, with an operating potential of 0.17 V. The electrocatalytic characteristics of the Pd-ZnO/GCE toward the oxidation of nitrite were measured in a phosphate-buffered solution (PBS, 0.1 mol/L, pH 7.0).

### 3.5. Pretreatment of Real Samples

Samples of sausage and pickles were obtained from a local supermarket and pretreated as follows: the samples were firstly homogenized with a motor-driven tissue grinder. Then, 2.0 g of homogenate was added to 80 mL of deionized water, and the sample solution was subjected to ultrasonication extraction for 30 min. Then, the abovementioned samples were heated in a water bath at 75 °C for 5 min and cooled to room temperature to make the protein precipitate. Then, the mixture was diluted with deionized water to 100 mL and centrifuged for 15 min at 10000 rpm. Finally, the supernatant was further filtered using a 0.22 µm membrane filter. The resultant filtrate was used as analytes for the optical method and electrochemical measurements.

The visible spectrophotometry was used as a reference method to measure nitrite in real samples. The measurement protocols were based on the national standards of China (GB5009.33-2016).

## 4. Conclusions

This work proposed a nitrite electrochemical sensor based on Pd-ZnO nanostructures. The Pd-ZnO nanocatalysts exhibited excellent electrocatalytic activities toward nitrite oxidation reaction, which was due to the high catalytic activity of the Pd NPs and the large active surface area of the ZnO support. Moreover, by tuning the solution pH, different surface morphologies of Pd-ZnO were fabricated and displayed different electrocatalytic activity levels, which could be ascribed to different electroactive surface areas of Pd-ZnO with different structures. Furthermore, the sensor based on a flower-like Pd-ZnO nanostructure had superior qualities, such as a wide linear range, high sensitivity, and a low detection limit, which were superior or comparable to those reported in other studies. Moreover, the Pd-ZnO sensing platform showed good selectivity, reproducibility, and stability for nitrite detection, which can used for real sample detection.

## Figures and Tables

**Figure 1 molecules-28-00099-f001:**
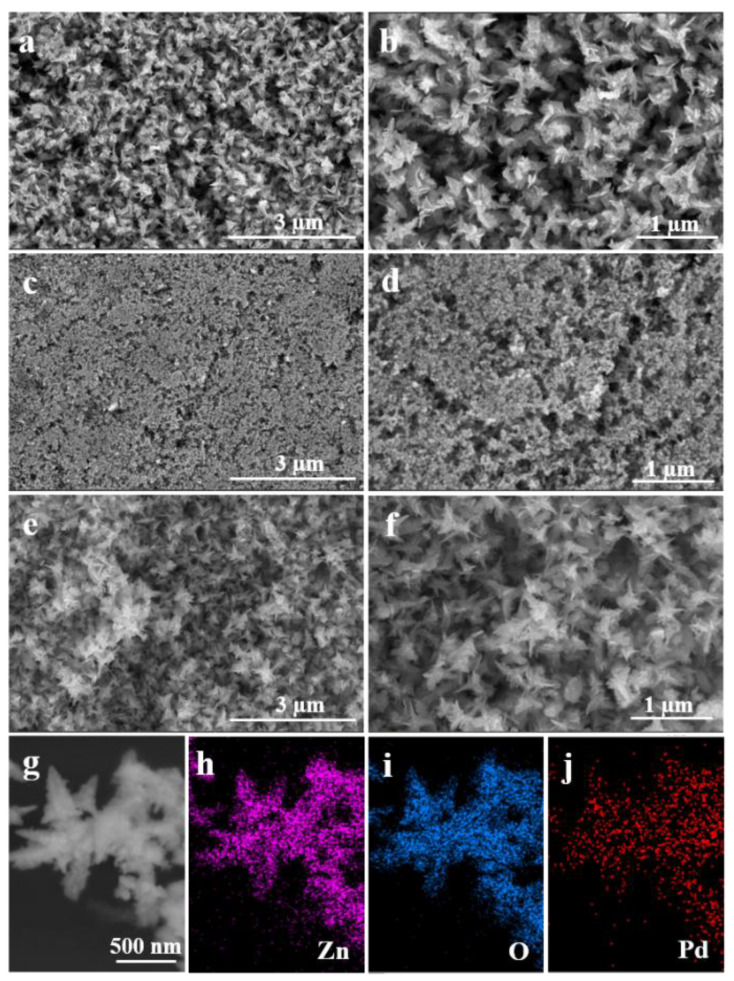
Typical SEM images of ZnO (**a**,**b**), Pd (**c**,**d**), and Pd-ZnO (**e**,**f**) nanostructures. (**g**) HAADF-STEM image of the Pd-ZnO nanocomposites, and (**h**–**j**) energy-dispersive X-ray (EDX) maps of Zn (**h**), O (**i**), and Pd (**j**), respectively.

**Figure 2 molecules-28-00099-f002:**
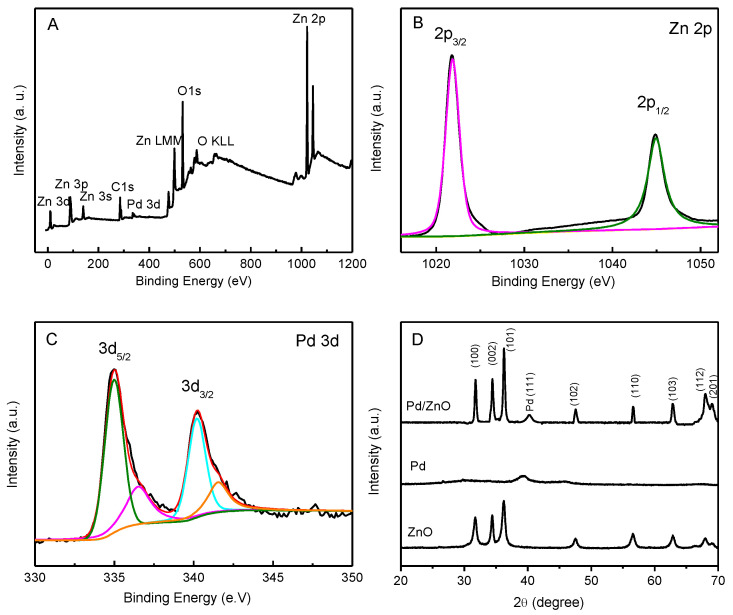
The survey XPS spectrum (**panel A**), the XPS spectra of Zn 2p (**panel B**), and Pd 3d (**panel C**) of Pd-ZnO nanostructures. **Panel D** shows the typical XRD patterns of ZnO, Pd, and Pd-ZnO nanostructures.

**Figure 3 molecules-28-00099-f003:**
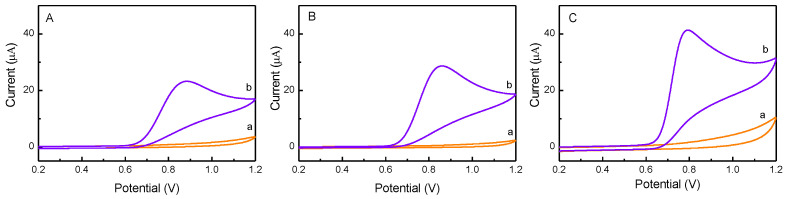
Cyclic voltammograms of ZnO/GCE (**A**), Pd/GCE (**B**), and Pd-ZnO/GCE (**C**) in the absence (curve a) and presence (curve b) of 0.5 mM nitrite in PBS (pH 7.0). The scan rate is 50 mV/s.

**Figure 4 molecules-28-00099-f004:**
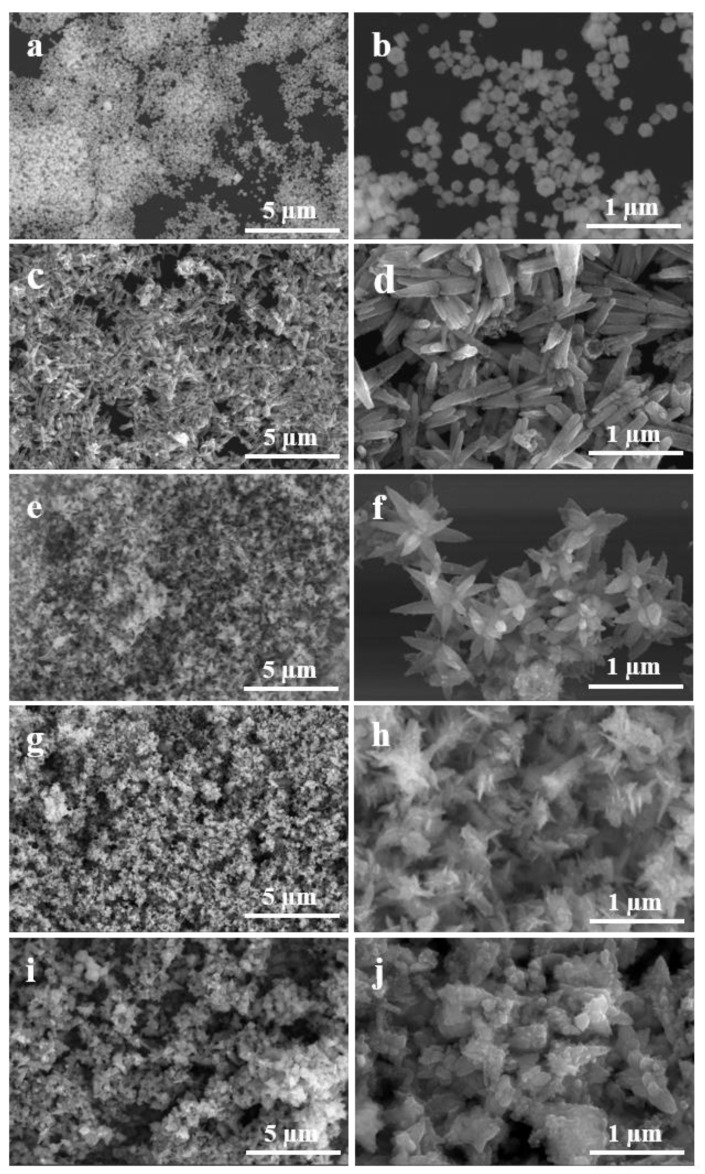
SEM images of the Pd-ZnO nanostructures fabricated at pH ≈ 6 (**a**,**b**), pH ≈ 7 (**c**,**d**), pH ≈ 8 (**e**,**f**), pH ≈ 9 (**g**,**h**), and pH ≈ 10 (**i**,**j**), respectively.

**Figure 5 molecules-28-00099-f005:**
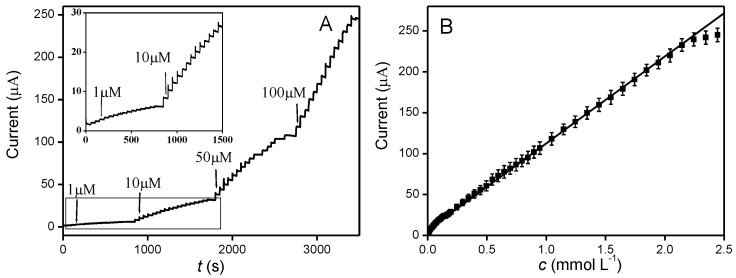
(**A**) Typical amperometric response of the Pd-ZnO/GCE at 0.78 V with the successive addition of nitrite in 0.1 M PBS (pH = 7.0). (**B**) Calibration curve of amperometric current versus the concentration of nitrite.

**Figure 6 molecules-28-00099-f006:**
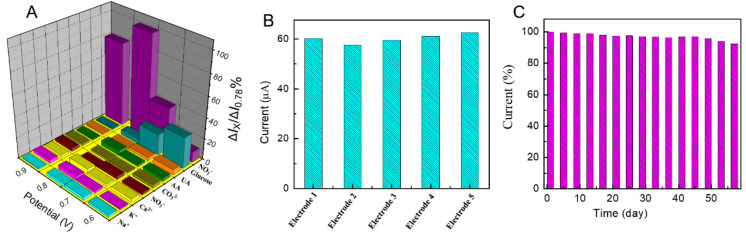
(**A**) The selectivity profile of the sensor in 0.1 M PBS at different potentials with 0.5 mM nitrite and other interferents. (**B**) Amperometric responses for 0.5 mM nitrite recorded with five different Pd-ZnO/GCE prepared using the same procedures. (**C**) Stability of the sensor stored under ambient conditions over two months using 0.1 M PBS at 0.78 V with 0.5 mM nitrite (*I*/*I*_0_).

**Table 1 molecules-28-00099-t001:** Comparison of the performance of the Pd-ZnO-modified electrode with other nitrite sensors based on different modified electrodes.

Electrode	Linear Range (µM)	Detection Limit (μM)	Ref
rGO-MoS_2_-PEDOT	1–1000	0.059	[16]
Poly(3ABA)	10–140	0.15	[17]
PMo_11_V/PDDA-rGO	0.125–1160	0.0028	[21]
f-ZnO@rFGO	100–3000	33	[22]
Pt-RGO/GCE	0.25–90	0.1	[23]
CDs-Au-N	0.1–2000	0.06	[25]
ERGO/AuNPs	1–6000	0.13	[26]
AuPd	2–4200	0.67	[27]
AuNPs/CP	1–100	0.093	[30]
CuOx/ERGO	0.1–100	0.072	[35]
NrGO	0.5–5000	0.2	[40]
Pd-ZnO	1–2350	0.2	This work

**Table 2 molecules-28-00099-t002:** Determination of nitrite concentration in sausage and pickles (*n* = 3).

Samples	Electrochemical Method (mg/kg)	Optical Method (mg/kg)	RSD (%)	Recovery (%)
Sausage	14.36	14.12	3.58	92.36
	25.74	26.26	3.97	105.78
	15.68	15.97	4.13	93.46
Pickles	8.79	8.23	4.56	108.56
	12.33	11.98	3.98	105.38
	16.98	17.06	3.45	97.88

## Data Availability

Not applicable.

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
