# Peer review of "Facile Controlled Synthesis of Pd-ZnO Nanostructures for Nitrite Detection"

_molecules, 2022, doi:10.3390/molecules28010099_

Round 1
Reviewer 1 Report
This paper is well-written and can attract the interest of readers. However, the reviewer feels minor corrections in the text. Please confirm the comments below.
1) L128: Why did the author determine the operating potential (modulation amplitude), 0.17 V, as most appropriate in the EIS measurement? It seems to be high for a liquid electrolyte and GCE.
2) Figure 2D: A small shoulder peak appeared in the lower diffraction angle, 2-theta, from the 002 reflection peak of ZnO. The authors did not refer to this peak. Did the Pd/ZnO include any impurity phase?
3) LL220-224: In Figure 3, the authors described the highest electrocatalytic activity of Pd-ZnO due to the highest peak current in the oxidation of the nitrite. However, the reviewer feels that the effect of the active surface areas in ZnO, Pd, and Pd-ZnO fixed on the GCE should be discussed.
4) Fig. S2: Do the prepared five Pd-ZnO samples have almost the same specific surface area to discuss their electrocatalytic activities from the oxidation current? Can the authors describe the difference in the oxidation peak potential among these samples?
Author Response
Dear Professor
Thank you very much for your comments on our above manuscript. The manuscript has been revised according to your comments. The followings are the detailed changes made in the revised manuscript. The changes made in the revised manuscript have been highlighted with yellow.
- Comment: L128: Why did the author determine the operating potential (modulation amplitude), 0.17 V, as most appropriate in the EIS measurement? It seems to be high for a liquid electrolyte and GCE.
Reply: The operating potential, 0.17V is the open circuit potential, which is measured by the CHI electrochemical work-station.
- Comment: Figure 2D: A small shoulder peak appeared in the lower diffraction angle, 2-theta, from the 002 reflection peak of ZnO. The authors did not refer to this peak. Did the Pd/ZnO include any impurity phase?
Reply: Thanks for your suggestion. The small shoulder peak from the (002) reflection peak of ZnO probably the noise. The X-ray diffraction (XRD) of the same Pd-ZnO nanocomposites were measured again, there is no any small peak around the 002 peak, please refer to the Figure 2D in the revised manuscript.
- Comment: L220-224: In Figure 3, the authors described the highest electrocatalytic activity of Pd-ZnO due to the highest peak current in the oxidation of the nitrite. However, the reviewer feels that the effect of the active surface areas in ZnO, Pd, and Pd-ZnO fixed on the GCE should be discussed.
Reply: Thanks for your suggestion. The reasons of the highest electrocatalytic performance of Pd-ZnO nanocatalyst were discussed in the revised manuscript, please refer to the L229-233.
- Comment: S2: Do the prepared five Pd-ZnO samples have almost the same specific surface area to discuss their electrocatalytic activities from the oxidation current? Can the authors describe the difference in the oxidation peak potential among these samples?
Reply: Thanks for your comment. The prepared five Pd-ZnO samples didn’t have the same specific surface area, and the surface area of the different Pd-ZnO samples were measured by BET, and the results were presented in Fig. S2 in the revised supporting information. Moreover, the difference in the oxidation peak potential among these samples were discussed, please refer to L286-288 in the revised manuscript.
Thank you very much for your assistance with this manuscript.
Sincerely yours
Yaojuan Hu
Reviewer 2 Report
This manuscript reported a nanoflower Pd-ZnO based nitrite sensor. Specifically, the synthetic method, morphological characterization, and electrocatalytical performance of the nanostructure were studied. Moreover, decent sensitivity, stability, and selectivity of the sensor were demonstrated. Overall, the paper is well written, but the following few points need to be addressed before publication:
1. line 164: Based on the EDS results in Figure 1, it is impossible to conclude that "Pd NPs were only uniformly dispersed on the surface of the ZnO". Since the Pd-ZnO nanostructure is synthesized via the co-reduction of Pd and Zn precursors, it is highly possible that Pd was also presented in the core of the nanostructure during nucleation, while some of Pd reside on the surface of the nanostructure that gives good electrocatalytic performance. Please remove this claim unless the authors have more evidence that Pd NPs were only on the surface. Moreover, a better way to make use of the Pd will be to synthesized the ZnO nanoflower support first, and then intentionally introduce the Pd on top of ZnO nanoflower.
2. section 3.3: The authors presented a few cases with different volume of NaOH added. It is clear that NaOH changes the solution pH and hence the final morphology of the nanostructure, but it would be much better to directly report the pH values of the synthetic solution instead of the volume of NaOH.
3. The authors claim the high surface areas associated with the nanoflower structure is the key for improved electrocatalytical performance, then it would be necessary to measure the specific surface area of different nanostructures shown in Figure 4 by BET to support the claim.
4. Figure 2d: The Pd(111) peak in Pd-ZnO shows a clear shift when compared with the Pd reference. Please provide the possible reasons in the main text.
Author Response
Dear Professor,
Thank you very much for your comments on our above manuscript. The manuscript has been revised according to your comments. The followings are the detailed changes made in the revised manuscript. The changes made in the revised manuscript have been highlighted with yellow.
- Comment: line 164: Based on the EDS results in Figure 1, it is impossible to conclude that "Pd NPs were only uniformly dispersed on the surface of the ZnO". Since the Pd-ZnO nanostructure is synthesized via the co-reduction of Pd and Zn precursors, it is highly possible that Pd was also presented in the core of the nanostructure during nucleation, while some of Pd reside on the surface of the nanostructure that gives good electrocatalytic performance. Please remove this claim unless the authors have more evidence that Pd NPs were only on the surface. Moreover, a better way to make use of the Pd will be to synthesized the ZnO nanoflower support first, and then intentionally introduce the Pd on top of ZnO nanoflower.
Reply: Thanks for your suggestion. The conclusion “Pd NPs were only uniformly dispersed on the surface of the ZnO” has been revised, please refer to line 171 in the revised manuscript. Moreover, the way of synthesized ZnO nanoflower support first, and then intentionally introduce the Pd on top of ZnO nanoflower has been investigated, and the results may be reported elsewhere in the future.
- Comment: section 3.3: The authors presented a few cases with different volume of NaOH added. It is clear that NaOH changes the solution pH and hence the final morphology of the nanostructure, but it would be much better to directly report the pH values of the synthetic solution instead of the volume of NaOH.
Reply: Thanks for your suggestion. The pH values of the synthetic solution have been reported, please refer to the L240-255, Figure 4, Figure S2, and Figure S3 in section 3.3 in the revised manuscript.
- Comment: The authors claim the high surface areas associated with the nanoflower structure is the key for improved electrocatalytical performance, then it would be necessary to measure the specific surface area of different nanostructures shown in Figure 4 by BET to support the claim.
Reply: Thanks for your suggestion. The specific surface area of different nanostructures has been measured by BET, and the results have been shown in Figure S3. The discussions have been presented in L 276-280 in revised manuscript.
- Comment: The Pd(111) peak in Pd-ZnO shows a clear shift when compared with the Pd reference. Please provide the possible reasons in the main text.
Reply: The possible reasons of the shift of Pd(111) peak have been provided in the revised manuscript, please refer to L193-195.
Thank you very much for your assistance with this manuscript.
Sincerely yours
Yaojuan Hu
Reviewer 3 Report
The paper entitled "Facile controlled synthesis of Pd-ZnO nanostructures for nitrite detection" (ijms-2042392) presents the synthesis of an electrode based on Pd-ZnO nanostructures for the oxidation of nitrite. The authors studied the relationship between morphological characteristics of the catalyst and the electrochemical performances of the electrode, comparing them with only ZnO, only NP of Pd and as a function of pH.
The work would seem interesting and suitable for publication with some small revisions, listed below:
1) different morphologies of the deposit are obtained by varying the pH of the solution (different additions of NaOH). The authors should explain The authors should explain what the effect of pH is on morphology
2) the samples were fabricated through a hydro-thermal method using DMF as the reduction agent. Authors should report the DMF reaction or a reference to such a reaction
3) Electrochemical Impedance spectroscopy (EIS) was investigated with the frequency range of 10-2 to 10-5 kHz, operated potential of 0.17 V. Is the frequency range correct? Why was that potential value chosen?
4) there are several grammatical and spelling errors (for example, which instead of which). Please check the text carefully
Author Response
Dear Professor:
Thank you very much for your comments on our above manuscript. The manuscript has been revised according to your comments. The followings are the detailed changes made in the revised manuscript. The changes made in the revised manuscript have been highlighted with yellow.
- Comment: different morphologies of the deposit are obtained by varying the pH of the solution (different additions of NaOH). The authors should explain what the effect of pH is on morphology.
Reply: Thank you for your suggestion. The effect of pH on the morphologies of the nanocomposites was explained in the revised, please refer to the L259-275.
- Comment: the samples were fabricated through a hydro-thermal method using DMF as the reduction agent. Authors should report the DMF reaction or a reference to such a reaction
Reply: Thank you for your suggestion. The control experiment without DMF was reported in the revised manuscript, please refer to the L154-159.
- Comment: Electrochemical Impedance spectroscopy (EIS) was investigated with the frequency range of 10-2 to 10-5 kHz, operated potential of 0.17 V. Is the frequency range correct? Why was that potential value chosen?
Reply: Sorry, the frequency range is of 10-2 to 105 Hz, the data has been revised in the revised manuscript, please refer to line 130. Thank you for your reminding. Moreover, the operated potential of 0.17V is the open circuit potential, which is measured by the CHI electrochemical work-station.
- Comment: there are several grammatical and spelling errors (for example, which instead of which). Please check the text carefully
Reply: Thank you very much. The text has been checked carefully.
Thank you very much for your assistance with this manuscript.
Sincerely yours
Yaojuan Hu
Round 2
Reviewer 2 Report
The revised manuscript is suitable for publication.